# Study on the Preferential Flow Characteristics under Different Precipitation Amounts in Simian Mountain Grassland of China

**Mingfeng Li [1], Jingjing Yao [2] and Jinhua Cheng [1,*]**

[1] Jinyun Forest Ecosystem Research Station, School of Soil and Water Conservation,
Beijing Forestry University, Beijing 100083, China; m17710660921@163.com

[2] Environmental Protection Research Institute of Light Industry, Research Center for Urban Environment,
Beijing Academy of Science and Technology, Beijing 100095, China; yaojing1989_lucky@163.com

* Correspondence: jinhua_cheng@126.com

**Abstract:** Understanding the response of preferential flow paths to water movement is an important topic in soil hydrology. However, quantification of the complicated distribution patterns of preferential flow paths remains poorly understood. Therefore, dye experiments were conducted to investigate preferential flow characteristics under three different precipitation amounts (20, 40 and 60 mm, numbered as the G20, G40 and G60, respectively) in Simian Mountain grassland, Chongqing province, China. O-ring statistics were used to analyze the spatial distribution characteristics and the spatial correlation of preferential flow paths. Results revealed that precipitation could promote dye tracer infiltration into deeper soils, reaching the maximum depth of 55 cm in G60. The number of preferential flow paths in G60 plots was 3.0 and 7.4 times greater than those of G40 and G20, respectively. Structural distribution of the preferential flow paths showed a gradually clumped pattern with the increase of precipitation, which was conducive to enhancing the correlation between preferential flow paths in each pore size range. These results could expand our understanding of the effects of precipitation on the characteristic of preferential flow paths in grassland, which is helpful to evaluate the water movement in the study area.

**Keywords:** precipitation amounts; preferential flow paths; grassland; spatial point pattern analysis

## 1. Introduction

Preferential flow is the fast, non-equilibrium flow of water infiltration in the soil that can reduce water and nutrient availability, threaten groundwater, and cause natural disasters such as avalanches, landslides, and mudslides [1,2]. The effect of preferential flow processes on hydrological processes has been widely discussed in the literature to predict soil solute transport and soil erosion [3–5]. An earlier study combined a process-based model with traditional tracer experiments to simulate and validate that the vertical and lateral flow paths play an important role in controlling infiltration and affecting surface and groundwater flows [6]. In other words, the spatial distribution of soil preferential paths directly influenced the water discharge and the process of rainfall recharging groundwater. Moreover, the spatial distribution of preferential paths reflected the connectivity of preferential paths along with the soil depth, which in turn affected the formation and development of preferential flow [7]. Consequently, determining the spatial pattern of preferential flow paths is integral to an understanding of the spatial heterogeneity and formation mechanism of preferential flow paths.

Existing research on preferential flow paths is limited because of the difficulty in determining the spatial position information in a horizontal soil profile [8]. A precise method of spatial analysis has yet to be established in this area of research. To date, the most straightforward way to study preferential

flow paths is to identify and label them by dye tracer tests [9]. This method uses the dye image to obtain direct information about the path number, dye depth and continuity of different water processes [10]. Its major limitation lies in the image analysis of the spatial structure characteristic of preferential flow paths, which are based on two-dimensional (2-D) soil sections. The computed tomography (CT) scanning technology has also been used to investigate three-dimensional (3-D) preferential flow paths by synthesising a series of 2-D images [11]. This method can provide the visual and quantitative information of the preferential flow paths' spatial variation. However, the approach is based on CT scanning, which is limited by its small sample size, prime instrument cost, long-time analysis, and high workload [4,12]. A few studies have applied water penetration curves and the Poiseuille equation to analyse the quantitative and morphological characteristics of soil macropore paths [13]. These studies have revealed the quantitative characteristics of preferential flow paths but has provided limited information regarding the distribution states [5]. O-ring statistics provide a quantitative analysis of the preferential flow paths of different size classes in horizontal soil profiles and thus determines the horizontal spatial distribution state of preferential flow paths [8]. This method can quantitatively show the spatial distribution characteristics of individuals through point pattern analysis.

Various factors have influenced the formation, distribution, and differentiation of the preferential flow, leading to complex and diverse research on the preferential flow [14]. Factors that affect preferential flow paths include soil types and structure, biological activities (channels of roots and earthworms), soil moisture content, and hydraulic conditions [15–17]. Soil types and structure have complex effects on preferential flow because of their spatial heterogeneities, which can directly change the hydraulic properties, quantities and distribution of soil macropores. Biological activities create complex channel systems that could serve as preferential flow paths, thereby further influencing the lateral and vertical movements of preferential flow [18]. The role of antecedent soil water in preferential flow may differ under different soil and macropore conditions [19]. These factors primarily affect the density and distribution of preferential flow paths, altering the soil macroporous structure and its connectivity through the soil, and consequently determining the scale and nature of the preferential flows. Hydraulic conditions such as rainfall intensity, duration, and total rainfall affects the momentum balance of water flow driven primarily by gravity [2]. Studies showed that increases in rainfall intensity can enhance preferential flow as a result of increased soil water pressure [20]. However, spatial changes of preferential flow under different amounts of precipitation have not been fully described and quantitatively tested, which are crucial to understanding the mechanism of preferential flow in different rainfall events.

The Simian Mountain is located in the transitional zone of the upper and middle Yangtze River in China, which is an important ecological barrier and a major water source in the middle and lower reaches of the Yangtze River. Studying the preferential flow in this region will help to fully understand the impact of preferential flow on the regional water cycle, as well as help to protect the regional water environment and ecological security. However, in previous studies more focused on the forest and farmland, grassland was limited. To address these limitations, we performed the dye experiments with different precipitation amounts in Simian Mountain grassland to gain an insight into the spatial pattern of preferential flow paths and investigate their response to precipitation using O-ring statistics. The present work focused on two aims: (1) examining the effects of different infiltration conditions on the numerical and spatial structure characteristics of preferential flow paths; and (2) analysing the spatial distribution characteristics and spatial correlation of the preferential flow paths with different influence radii.

## 2. Materials and Methods

### 2.1. Field Site

The experiments were conducted in the natural grassland soil of Simian Mountain in Chongqing (28°31′–28°43′ N, 106°17′–106°30′ E), China. The site has a humid subtropical monsoon climate,

a mean annual precipitation of 1522.3 mm and a mean annual temperature range of 13.6–18.4 °C. The altitude ranges from 500 m to 1780 m, and the soil type characteristics are mainly yellow earth and yellow-brown earth. The dominant plants are *Stenoloma chusanum Ching, Dicranopteris chinensis Tagawa, Carex siderosticta Hance*, and *Dicranopteris dichotoma Berhn.*

## 2.2. Experimental Design

In this study, three rainfall precipitation amounts of 20 mm (moderate rain), 40 mm (heavy rain), and 60 mm (rain storm) were prepared and numbered correspondingly as G20, G40, and G60. Nine plots were established for the three types of rainfall precipitation, with each type having three replicates. In each plot, based primarily on double-ring infiltration, two rectangular steel frames with an inner one of 60 cm (length) × 60 cm (width) × 50 cm (height) and an outer one of 80 cm × 80 cm × 50 cm were concentrically embedded into the soil to a depth of 30 cm. In order to facilitate the subsequent image analysis, we selected a rectangular steel frame instead of a circular one in the experiment, and they have identical effects [21]. After embedding, the soil within 5 cm was compacted to prevent dye tracer infiltration along with the frames. With the use of a sprayer with nozzles (GP1000, Baoding, China) operated above the centre of the frame, brilliant blue FCF (4 g L$^{-1}$) was sprayed slowly on the soil surface as a dye tracer to detect preferential flow paths. The dye solution was transported to the sprayer by a constant flow pump (BT100-02, Baoding, China). The usage amount of dye tracer is shown in Table 1. During the spraying experiment, an equal depth of water was sprayed at the soil between two steel frames to avoid the lateral infiltration of the brilliant blue solution. After 24 hours, the frames' soil edges were disturbed when the frames were moved, so that only the core part of the soil was analysed. The core region of the dye area (50 cm × 50 cm) was excavated in 12 horizontal, 5 cm-deep cross-sections. Each horizontal cross-section was vertically photographed with a digital camera (500D, Canon, Japan). The schematic layout of the dyed soil profiles is shown in Figure 1. Next to each plot, we excluded the litter layer and collected soil samples of approximately 500 g in triplicate at 10 cm-deep intervals along the soil depth to measure mass moisture content, bulk density, total porosity, soil mechanical composition, pH and organic matter content (Table 2).

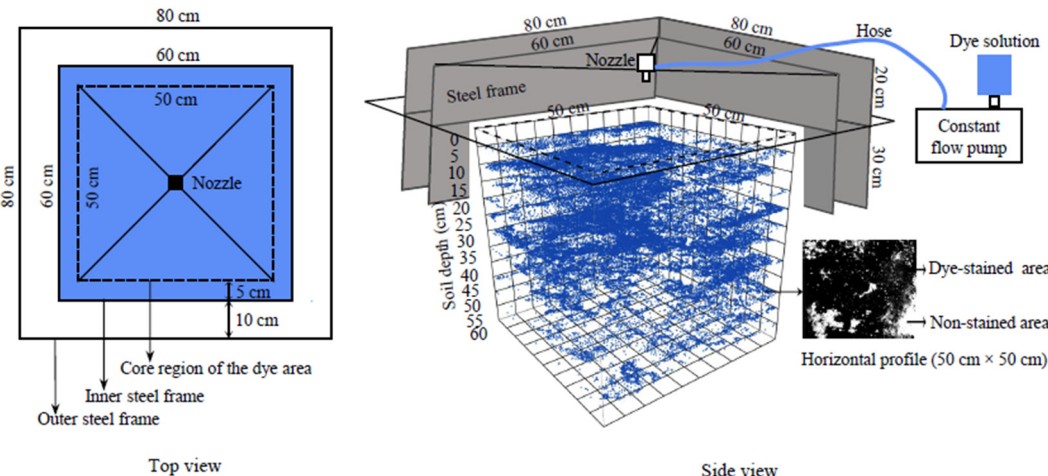

**Figure 1.** Schematic layout of the dyed soil profiles at one sampling position.

**Table 1.** Usage amount of dye tracer.

| Plot | Spray Volume (L) | Spray Rate (mm h$^{-1}$) | Spray Duration (h) |
|------|------------------|--------------------------|--------------------|
| G20 | 7.2 | 6.35 | 3.15 |
| G40 | 14.4 | 11.98 | 3.34 |
| G60 | 21.6 | 24.19 | 2.48 |

<div align="center">

**Table 2.** Physical properties of soil.

</div>

| Soil Depth (cm) | Moisture Content (%) | Bulk Density (g cm$^{-3}$) | Total Porosity (%) | Sand Content (%) | Silt Content (%) | Clay Content (%) | pH | Organic Matter Content (g kg$^{-1}$) |
|---|---|---|---|---|---|---|---|---|
| 0–10 | 42.43 ± 2.79 | 1.00 ± 0.07 | 56.67 ± 3.13 | 69.22 ± 7.21 | 27.19 ± 6.00 | 3.58 ± 0.21 | 4.34 ± 0.05 | 6.47 ± 0.21 |
| 10–20 | 31.71 ± 1.47 | 1.09 ± 0.06 | 49.76 ± 3.07 | 61.59 ± 3.72 | 32.39 ± 3.50 | 6.02 ± 0.23 | 4.40 ± 0.06 | 3.45 ± 0.17 |
| 20–30 | 29.44 ± 0.36 | 1.11 ± 0.08 | 47.69 ± 1.13 | 54.34 ± 5.66 | 37.79 ± 4.08 | 7.87 ± 1.96 | 4.41 ± 0.04 | 3.38 ± 0.04 |
| 30–40 | 30.99 ± 1.10 | 1.11 ± 0.08 | 50.27 ± 2.30 | 48.43 ± 4.81 | 44.58 ± 4.20 | 6.99 ± 0.65 | 4.48 ± 0.08 | 3.37 ± 0.11 |
| 40–50 | 35.82 ± 2.51 | 1.12 ± 0.04 | 51.76 ± 0.89 | 48.83 ± 3.44 | 42.14 ± 2.52 | 9.03 ± 1.03 | 4.39 ± 0.02 | 2.77 ± 0.11 |
| 50–60 | 34.88 ± 3.14 | 1.19 ± 0.05 | 49.72 ± 3.05 | 41.41 ± 2.46 | 47.84 ± 2.49 | 10.75 ± 0.42 | 4.43 ± 0.07 | 1.75 ± 0.13 |

<div align="center">

Note: Data in the table are the average value ± standard deviation.

</div>

### 2.3. Image Analysis

Image analysis in this work was completed in four steps. First, the horizontal section dyeing image was corrected and tailored to the ARCMAP 10.2 software (Environmental Research Systems Institute, Redlands, CA, USA) to a size of 50 cm × 50 cm, which was equivalent to 500 × 500 pixels (Figure 2a). Second, Adobe Photoshop CS3 (Adobe Systems, San Jose, CA, USA) was used to distinguish dyed areas from non-dyed areas through the functions of brightness, contrast, hue, and saturation change. The colour replacement function was then used to replace the dyed areas with black and non-dyed areas with white (Figure 2b). Third, the erode and dilate functions of Image-Pro Plus 6.0 software was used to reduce the small number of noise points produced by the two previous steps to simplify the image parsing process. The Bitmap Analysis function of Image-Pro Plus 6.0 software was used to convert the image into numerical matrices, consisting of only two numerical values of 0 and 255. The non-dyed areas are composed of many 255 pixel values and the dyed areas are composed of many 0 pixel values (Figure 2c). Fourth, the classification and count functions in Image-Pro Plus 6.0 software were used to distinguish the size classes and spatial position messages of the preferential flow in the dyed areas. The watershed function was set to divide the connected dyed areas into independent individuals (Figure 2d). The area of each dyed area was the number of pixels occupied by plaque, which were approximated as rings, and the radius of influence was calculated based on the stained area (Figure 2e). Then, a data file was generated containing the radius of dyed areas and the coordinates at the centre point of the dyed areas (Figure 2f). This study classified radii of preferential paths into four classes: 1.0–2.5, 2.5–5.0, 5.0–10.0, and >10.0 mm. An artificial statistical method was used to get the plant roots' positions in the horizontal images with the naked eye after the first step of image correction.

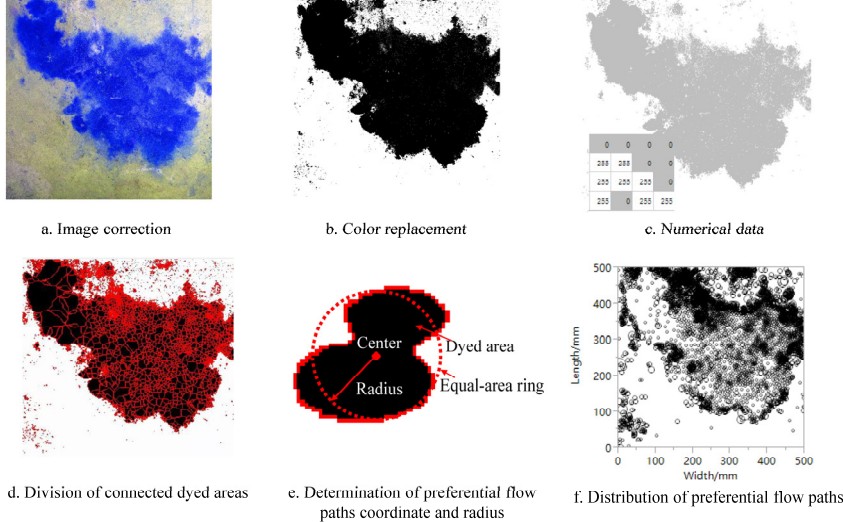

<div align="center">

**Figure 2.** Image Analysis.

</div>

*2.4. Spatial Point Pattern Analysis*

O-ring statistics, based on Ripley's K-function and the mark correlation function, were used to analyse the spatial patterns. The O-ring statistics replace the circles with rings as a statistical tool to eliminate the bias of accumulative measures using the probability density function.

O-ring statistics include both univariate and bivariate statistics. The statistics were used to analyse the spatial distribution pattern of the preferential flow path with the influence radius (Equation (1)). Bivariate statistics were used to analyse the spatial association of the preferential flow paths with two different influence radii (Equation (2)). Full details of O-ring statistics processing followed the method described by Thorsten [22].

$$O_{11}(r) = \frac{\sum\limits_{i=1}^{n} Points[R_i^w(r)]}{\sum\limits_{i=1}^{n} Area[R_i^w(r)]}, \tag{1}$$

where $O_{11}(r)$ is a univariate O-ring statistic, $n$ is the number of preferential flow paths in the observed area, $R_i^w(r)$ is the ring with radius $r$ and ring width $w$ centred in the $i$th point, $Points[R_i^w(r)]$ is the number of preferential flow paths within the ring and $Area[R_i^w(r)]$ is the area of the ring.

$$O_{12}^w(r) = \frac{\frac{1}{n_1}\sum\limits_{i=1}^{n_i} Points_2[R_{1,i}^w(r)]}{\frac{1}{n_1}\sum\limits_{i=1}^{n_i} Area[R_{1,i}^w(r)]}, \tag{2}$$

where $O_{12}^w(r)$ is bivariate O-ring statistic, subscripts 1 and 2 refer to pattern 1 and pattern 2 respectively, pattern 1 is the pattern with the smaller influence radius, pattern 2 is the pattern with the larger influence radius, $n_1$ is the number of preferential flow paths in pattern 1, $R_{1,i}^w(r)$ is the ring with radius $r$ and ring width $w$ centred in the $i$th point in the pattern 1, $Points_2[R_{1,i}^w(r)]$ is the number of preferential flow paths within the ring in the pattern 2, and $Area[R_{1,i}^w(r)]$ is the area of the ring.

Point pattern analysis was performed using Programita 2014 software (freely available at www.programita.org). The initial input data was a DAT file exported by image analysis, containing the coordinates and radii of the preferential flow paths. It took 1/5 (100 mm) of the sample edge length as the research scale and 1 mm as the step size. To obtain meaningful predictions, we selected a different null model separately for the univariate and the bivariate pattern. For univariate point patterns, if the scatterplot was uniformly distributed, then the complete spatial randomness (CSR) null model was used; if the distribution of sample points showed significant spatial heterogeneity, then the heterogeneous Poisson process (HPP) null model was used. For bivariate patterns, if the sample points representing the two variables were independent of each other, then the independence null model was used; if influenced by the heterogeneous environment, then the random labelling (RL) null model was used. A total of 19 Monte Carlo simulations were performed to get 95% confidence intervals (i.e., upper and lower package traces). On a certain spatial scale, for univariate statistics, if the $O_{11}(r)$ value is greater than the upper envelope trace, then the distribution is aggregated; when falling into the upper and lower envelope trace, the distribution is random; when it is less than the lower envelope trace, the distribution is uniform. For bivariate statistics, if the $O_{12}(r)$ value is larger than the upper envelope trace, then the two are positively correlated in space; when within the upper and lower envelope trace, the two are independent of each other; when less than the lower envelope traces, the two are negatively correlated in space.

## 3. Results and Discussion

### 3.1. Quantities and Locations of Preferential Flow Paths Under Different Infiltrations

For the movement characteristics of water in the active flow field, the topsoil (0–10 cm) was stained greater than 80% with brilliant blue dye, but relevant information about the preferential flow path in the surface soil could not be accurately extracted [23]. Therefore, to minimise the influence of matrix flow dyeing on the spatial structure analysis of the preferential flow path and to ensure the comparability of the research, only the preferential flow path of 10–60 cm soil depth in each plot was analysed. On the basis of the detection results, the preferential flow paths of the horizontal dyed image could be divided into four size classes: 1.0–2.5, 2.5–5.0, 5.0–10.0, and >10.0 mm.

The statistics revealed that the amount of infiltration directly affected the depth of dye infiltration. The staining depths of G20, G40, and G60 were 40, 50, and 55 cm, respectively, suggesting that the staining depth increased with infiltration (Figure 3). The results were consistent with earlier studies that soil–water movement depends strongly on rain intensity [20]. In addition, the number of preferential flow paths in the G60 plots was 3.0 and 7.4 times greater than that of G40 and G20, respectively (Figure 3). This result showed that the number of preferential flow paths increased concurrently with increasing infiltration intensity, and the effective hydraulic conductivity of the soil increased significantly. Specifically, the quantities of preferential flow paths in the three plots showed W-shaped small serrated patterns with increasing soil depth, and the peaks mainly appeared in the 10–35 cm soil depth (Figure 3), indicating more open and connected preferential flow path structures in this soil depth range. The major reason for this was that the 10–40 cm soil depth was consistent with the observed root growth range in actual observations. Evidence showed that over 80% of roots were distributed in the 0–40 cm soil layer for each root size range (Table 3). The denser shallow root system of grassland promoted the formation of preferential flow paths in shallow soil and effectively conducted soil moisture [7]. However, the peak of the count of preferential flow paths under G20 appeared at 30 cm soil depth and varied with a small fluctuation process. The first peak of the number of preferential flow paths under G40 occurred at 15 cm soil depth, followed by peaks at 25, 35, and 45 cm, respectively. In G60, the first peak of the preferential flow path was observed in the 20 cm soil depth and the second peak in the 30 cm soil depth, while the number of preferential flow paths in the soil depth below 30 cm showed a slowly decreasing trend (Figure 3). These outcomes suggest that increased infiltration can change the location of preferential flow paths and promote the differentiation of preferential flow paths [24].

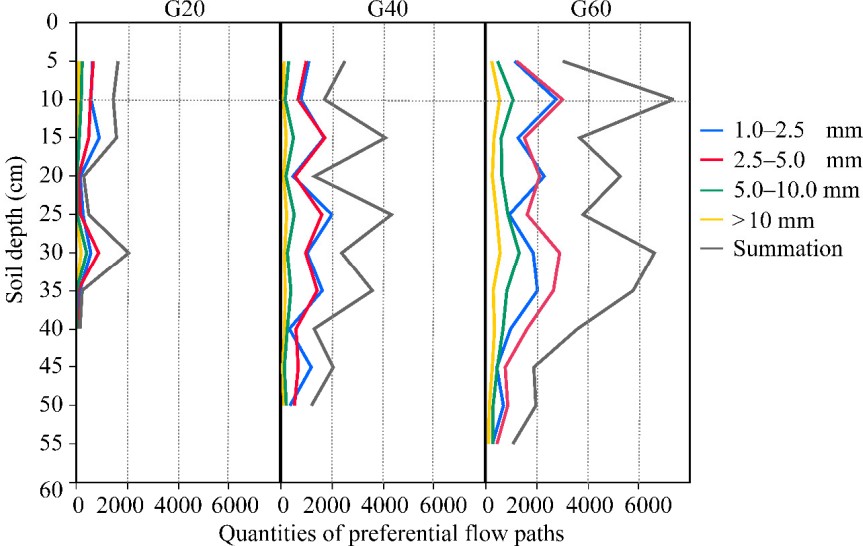

**Figure 3.** Number of preferential flow paths in different soil layers under different amounts of infiltration.

**Table 3.** Root length density (m m$^{-1}$).

| Soil Depth (cm) | Root Diameter | | | | |
|---|---|---|---|---|---|
| | <1 mm | 1–3 mm | 3–5 mm | 5–10 mm | >10 mm |
| 0–5 | 907.33 ± 54.02 | 166.93 ± 49.83 | 186.93 ± 121.08 | 38 ± 27.94 | 4 ± 3.27 |
| 5–10 | 179.87 ± 50.43 | 25.33 ± 22.31 | 3.87 ± 5.47 | 12 ± 8.64 | 2.83 ± 4 |
| 10–15 | 94.56 ± 59.12 | 35.12 ± 21.61 | 1.39 ± 1.96 | 3.97 ± 5.62 | 0 |
| 15–20 | 100.88 ± 26.81 | 0 | 6.24 ± 6.54 | 0 | 0 |
| 20–25 | 48.19 ± 13.67 | 6.21 ± 7.79 | 0 | 1.04 ± 1.47 | 0 |
| 25–30 | 41.2 ± 9.41 | 3.73 ± 5.28 | 0 | 1.33 ± 1.89 | 0 |
| 30–35 | 57.65 ± 60.58 | 32.8 ± 41.4 | 2.72 ± 2.01 | 0.69 ± 0.98 | 0 |
| 35–40 | 124.45 ± 97.23 | 41.87 ± 32.77 | 23.81 ± 26.16 | 13.6 ± 10.79 | 0 |
| 40–45 | 40.29 ± 20.69 | 22.32 ± 20.97 | 10.37 ± 11 | 3.89 ± 2.76 | 0.72 ± 1.02 |
| 45–50 | 14.4 ± 10.37 | 1.07 ± 1.51 | 0 | 5.6 ± 2.26 | 0.69 ± 0.49 |
| 50–55 | 8.77 ± 9.22 | 1.63 ± 1.34 | 0 | 3.41 ± 1.28 | 0 |
| Total root length density | 1617.59 | 337.01 | 235.33 | 83.53 | 8.24 |

Statistics on the number of preferential flow paths with different aperture size ranges are presented in Figure 3. In G20, the number of preferential flow paths for different aperture size ranges at 10–25 and 25–40 cm soil depth were 1.0–2.5, 2.5–5.0, 5.0–10.0, and >10.0 mm and 2.5–5.0, 1.0–2.5, 5.0–10.0, and >10.0 mm, respectively. In G40, there was no significant difference in the number of preferential flow paths in the 1.0–2.5 and 2.5–5.0 mm pore size ranges, followed by 5.0–10.0 and >10.0 mm at 10–60 cm soil depth. In G60, the number of preferential flow paths for different pore size ranges in descending order at the 10–60 cm soil depth was 2.5–5.0, 1.0–2.5, 5.0–10.0, and >10.0 mm. These results demonstrated that at the same depth, G20, G40, and G60 mostly showed greater numbers of preferential flow paths in the small pore size ranges (1.0–2.5 and 2.5–5.0 mm) than in the large pore size ranges (5.0–10.0 and >10.0 mm). This variation may be related to the number of plant roots of different diameter classes in the soil [25–27]. An investigation into the root systems in the Simian Mountain grassland revealed that more than 90% of root distribution is in the range of < 5.0 mm diameter classes (Table 3). The number of preferential flow paths in the larger pore size range also increased with the increase of infiltration. The number of preferential flow paths in the 2.5–5.0 mm range at G60 was higher than that in the 1.0–2.5 mm range, indicating that with the increase of infiltration water, the preferential paths of large pore size were stimulated, thus increasing the downward transport of soil water.

*3.2. Spatial Distribution Patterns of Preferential Flow Paths in Different Infiltrations*

The spatial distribution of preferential flow paths in soil reflects the up-and-down connectivity of preferential flow paths along with soil depth, which influences the formation and development of priority flows. Figure 4 presents the changing state of the spatial distribution of preferential flow paths at different scales for each radius of influence under different infiltrations, revealing the degree of up-and-down connectivity of the preferential flow paths. Figure 5 displays the location of preferential flow paths at different soil depths for different infiltration volumes, reflecting their overall distribution in the soil space.

The combination of Figures 4 and 5 reveals that the distribution patterns of preferential flow paths in the 1.0–2.5 and 2.5–5.0 mm pore diameter ranges were almost identical in G20, G40, and G60. Moreover, most of these patterns showed a clumped distribution at 10–60 cm soil depth, and the peaks were all in the 0–30 mm scale range. This finding was consistent with previous findings that preferential flow paths with small size classes displayed similar distribution tendencies and that preferential flow paths in this aperture range were well connected in the vertical soil profile [11]. It also indirectly indicated that the distribution patterns of preferential flow paths with a semi-diameter of 1.0–5.0 mm tended to have a highly preferential transport capability and hydraulic conductivity

compared with other distribution patterns of preferential flow paths of different sizes. However, in G20, the variations of preferential flow paths in the 20 and 35 cm soil layers were complicated, and the $O_{11}(r)$ function curve fluctuated up and down near the upper envelope line. This result may be attributed to the fact that the number of preferential flow paths in the 20 and 35 cm soil layers were at a minimum (Figures 3 and 5). The distribution patterns of preferential flow paths in the 5.0–10.0 and >10.0 mm pore size ranges varied considerably across different infiltrations. In G20, the curves failed to reflect the distribution state of the preferential flow paths because of the small amounts of preferential flow paths in the 5.0–10.0 and >10.0 mm aperture ranges. In G40, the distribution pattern curve of 5.0–10.0 mm preferential flow paths first increased, then decreased and fluctuated up and down near the upper boundary of the envelope with the increase of the distance scale, indicating that the distribution of preferential flow paths gradually changed from clumping to a random distribution. Meanwhile, the distribution of >10.0 mm preferential flow paths was mainly random. In G60, the distribution of preferential flow paths for each pore range exhibited a distinct clumped distribution. Collectively, these results indicated that the entire preferential flow path spatial structure tended to feature from the random to the clumped distribution patterns with the increase of infiltration and easily formed the upper and lower connectivity of paths.

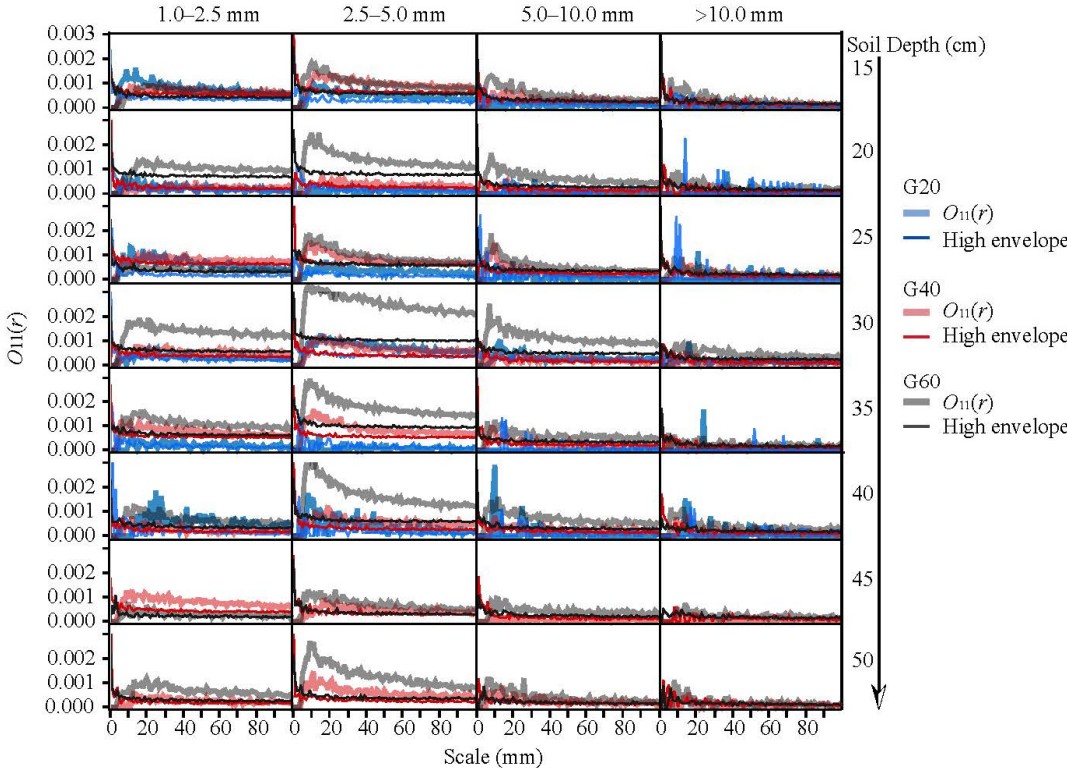

**Figure 4.** Change of horizontal distribution pattern function $O_{11}(r)$ with scale at different soil depths.

Furthermore, Figure 5 demonstrates that a few preferential flow paths existed in the G20, and the distribution of preferential flow paths, without being apparent, varies along with the depth. Only a few preferential flow paths were generated in the 30 cm layer through the gathering of water and the occurrence of lateral infiltration. In G40, the distribution area ratio of preferential flow paths increased in the 15, 25, 35, and 45 cm layers compared to that in the previous soil layer, showing that the accumulation and lateral infiltration of water flow in these soil layers facilitated the formation and expansion of the preferential flow paths. Additionally, the distribution of the preferential flow paths changed significantly in the 35 to 40 cm soil layers, indicating that the preferential flow paths had a certain degree of lateral offset. In G60, the distribution area ratio of preferential flow paths decreased with increasing soil depth, while the distribution of the preferential flow paths varied insignificantly.

This result indicates that the clumped distribution of preferential flow paths increased the vertical transport of water and solutes as well as decreased the lateral transport.

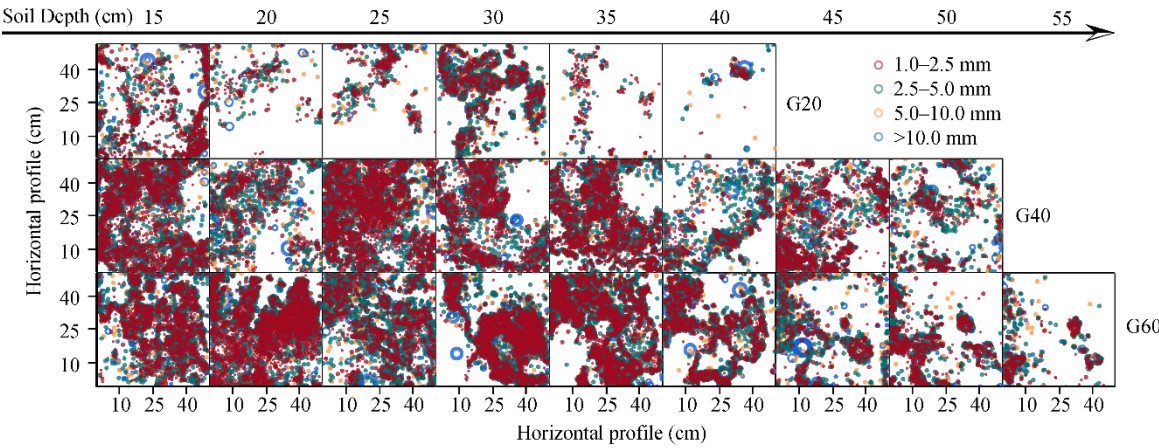

**Figure 5.** Distribution map of preferential flow paths in horizon dye profiles. Note: Values in the figure are the distribution area ratio (%). Distribution area ratio is the ratio of the distribution area of preferential flow paths to the total area.

### 3.3. Spatial Association of Preferential Flow Paths Under Different Infiltrations

To further reveal the spatial structure characteristics of soil preferential flow paths, spatial correlation analyses of the spatial location relationships between different preferential flow paths for different pore size ranges were conducted. The spatial association of paths of different size classes with G20, G40, and G60 plots is shown in Figure 6.

The spatial association of preferential flow paths between 1.0–2.5 and 2.5–5.0 mm size classes exhibited significant positive correlations in 10–60 cm soil layers under three different infiltration treatments. Additionally, the value of the $O_{12}(r)$ function increased with the increasing amount of water infiltrated. The formation and distribution of preferred paths between 1.0–2.5 and 2.5–5.0 mm pore sizes were shown to influence each other, and their degree of correlation was enhanced by the increasing water infiltration. There were mostly significant positive correlations between the preferential flow paths in the 1.0–2.5 and 5.0–10.0 mm pore size ranges, except at 20 and 35 cm soil depths in G20 and 20 and 45 cm soil depths in G40, where the value of $O_{12}(r)$ was close to or coincided with the upper bound. Analysis of spatial correlations between preferential flow paths with apertures of 1.0–2.5 and >10.0 mm, 2.5–5.0 and 5.0–10.0 mm, 2.5–5.0 and >10.0 mm, and 5.0–10.0 and >10.0 mm found mostly insignificant correlations in the G20 and mostly significant positive correlations in G60. Different preferential flow paths also showed no relationship, except for the paths of 2.5–5.0 and 5.0–10.0 mm in G40. These observations suggested that an increase in the amount of water infiltration was conducive to the enhancement of the degree of correlation between preferential flow paths in each pore size range. In other words, the preferential flow network expands when the degree of saturation increases.

The expanding network of active macropores leads to less resistance to the overall flow in the domain and access to increased volumes of the flow domain [28]. In this study, when the infiltration volume was 20 mm, fewer preferential flow paths failed to form a path network, and their distribution was random and unconnected. This result may be attributed to the fact that when the infiltration water volume was low, it was insufficient to meet the capacity of the preferential flow paths to transport the water downward [29]. When the upper infiltration water volume increased to 40 mm, the preferential flow paths increased, resulting in the vertical and lateral formation of the preferential flow network. By contrast, this preferential flow network was mainly composed of preferential flow paths in the small pore size ranges (1.0–2.5 and 2.5–5.0 mm) rather than the preferential flow paths in the large pore size ranges (5.0–10.0 and >10.0 mm) with random distribution and limited connection. This result

was largely explained by the capacity of the vertical preferential flow paths to transport water being exceeded, the blockage of the preferential flow paths, and the occurrence of lateral flow along the horizontal direction, thereby promoting the formation of new preferential flow paths [30]. When the infiltration volume was further increased to 60 mm, the preferential flow paths at each size level were clustered and connected, which in turn increased the infiltration uniformity and vertical infiltration intensity and promoted the movement of water to deeper layers [28]. Thus, it can be seen that under different infiltration conditions, the preferential flow paths show spatial heterogeneity in both vertical and horizontal directions, that is, they have 3-D spatial variability.

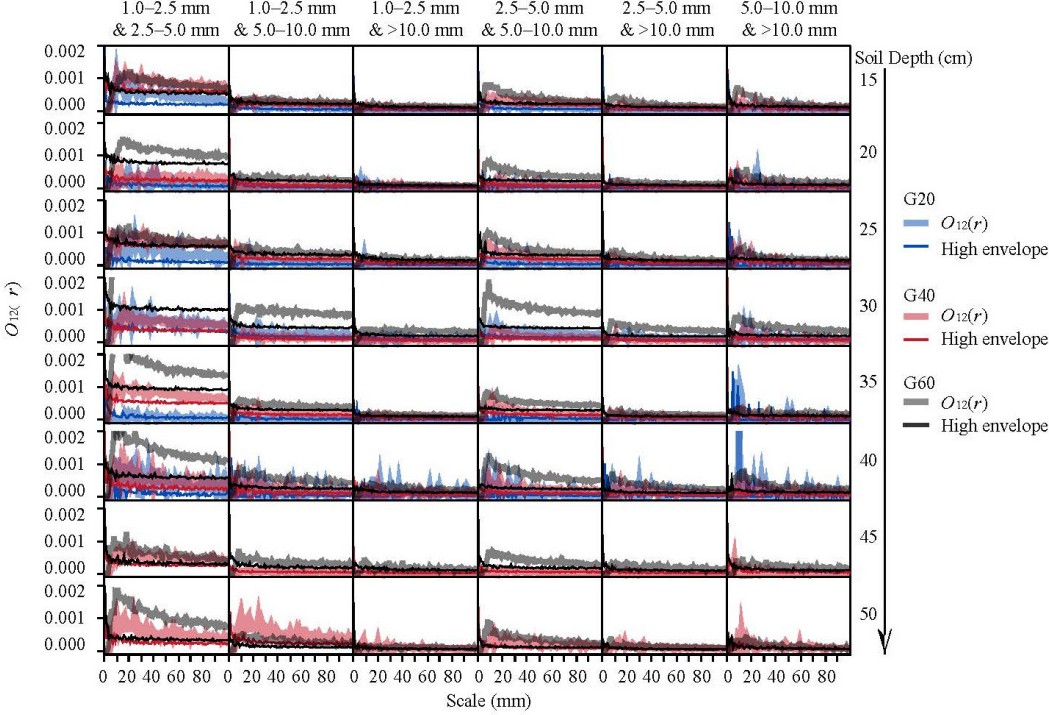

**Figure 6.** Change of spatial association function $O_{12}(r)$ with scale at different soil depths.

## 4. Conclusions

In summary, this study revealed the responses of spatial distribution and spatial association of preferential flow paths to different water infiltration conditions in the Simian Mountain grasslands of Chongqing, which may be of significance in exploring the characteristics of preferential flow movement. Our results indicated the important contribution of infiltration intensity to the quantities, spatial distribution, and spatial association of preferential flow paths. The increase in precipitation amounts was conducted to the preferential infiltration of water into deeper soil. Under moderate rain, preferential flow mainly followed the native preferential flow paths to infiltration, and less preferential flow paths were divided. The lateral offset of preferential flow paths was more obvious under heavy rain. Under rain storm, the concentrated distribution pattern resulted in highly continuous and effective preferential flow paths, and the large localised hydraulic gradients could lead to groundwater contamination and subsurface erosion. The findings of this study could provide a reference for future research on the application of preferential flow paths in hydrology processes in the Simian Mountain grasslands. Furthermore, spatial point pattern analysis may be helpful to quantitatively analyse the characteristics of the horizontal spatial distribution of preferential flow paths and thus accurately evaluate the spatial structure characteristics and development degree of preferential flow paths.

**Author Contributions:** Conceptualization, M.L.; methodology, M.L. and J.Y.; software, M.L.; validation, M.L.; formal analysis, M.L. and J.C.; investigation, M.L.; resources, M.L. and J.C.; data curation, M.L.; writing—original draft preparation, M.L., J.C. and J.Y.; writing—review and editing, M.L. and J.Y.; visualization, M.L.; supervision,

M.L.; project administration, M.L.; funding acquisition, M.L. and J.C. All authors have read and agreed to the published version of the manuscript.

**Funding:** This research was funded by the National Science and Technology Major Project (2018ZX07101005) and the National Natural Science Foundation of China (32071839).

**Acknowledgments:** Many thanks to the Forest Management Station of Simian Mountain for offering accommodations and supporting field experiments. We also gratefully acknowledge the editor and reviewers.

**Conflicts of Interest:** The authors declare no conflict of interest.

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
