# Peer review of "Study on the Preferential Flow Characteristics under Different Precipitation Amounts in Simian Mountain Grassland of China"

_water, doi:10.3390/w12123489_

Round 1

Reviewer 1 Report

This study presents an interesting analysis of the spatial patterns of preferential flow paths with a set of experiment data. The experiment seems overall well-conducted, and the application of the O-ring statistics indeed provided insights in to the experiment data. However, the presentation of the paper needs a lot of improvement to justify the content as a solid scientific journal paper.

First of all, I suggest a major revision of the abstract. 1) The audience would not know where Simian mountain or Chongqing province is, so it's better to include a proper reference of a your study area; at lease mention that it's in China (I assumed so). 2) Likewise, the code name (e.g., G20) would be meaningless to the readers of this paper before they actually read the whole paper. Please include proper reference of the experiment sites/setup. 3) The abstract mentions Ripley's K function but the authors actually applied O-ring statistics. This would confuse the audience. 4) In general, the abstract is not well-written in a sense that one could get the overall idea of the paper without reading it. Vague descriptions such as "summation of preferential flow paths" confuses people because the readers do not know how  the experiment was conducted, and do not know whether it's summation of cell count, or total size or macropore, etc. 

Line 31: "perfectly predict" is a very strong claim, and I'm not sure any study would claim that there's a way to perfectly predict preferential flow. If the authors believe otherwise, please elaborate.

"Soil Texture": this term is used in line 41 and line 81. I would suggest elaboration of what this term means in this paper's context. soil texture sometimes means different things in the field of geology, hydrogeology, soil mechanics, and foundation engineering. Thus, depending on the background of the audience, one may be very confused by the description of soil texture in line 81.

Research objective definition: The research objective seems to be in line 49-50. The next paragraph talks about position determination, but the present study does not focus on that part at all. I suggest a reorganization of the last 2 paragraphs in the introduction. It is also not introduced why it's important to study the spatial pattern of preferential flow paths.

Section 2.2: Please specify the precipitation experimental setup. What was the duration, and how was uniform precipitation over the plot achieved, etc. Please specify volumetric or gravimetric moisture content. Also, since double-ring infiltration test is mentioned in the paper, please justify the use of rectangular frames.

Line 118: Please elaborate on the calculation of the path sizes. Please also make it clear how "a path" is determined, and how the corresponding path size is determined. These are all very important aspects of your study and play an important part in the results. Without these details it's difficult for the readers to correctly interpret the results (e.g., how do we interpret "summation of flow path? cell count in the image analysis? total additive path size? path number count?).

Line 123: O-ring statistics and K function are actually suited for different research question. The authors should clarify the research question and justify the use of O-ring statistics. It is quite confusing to mention one method in the abstract and introduction, but apply another method in the rest of the study.

Equations 1 and 2: Please check for typos as sometimes there's a subscript in the texts but not in the equation. The bigger issue is that these two equations are poorly explained. It's not clear how "n" and "Points[R(r)]" are different based on the description. It is also not clear from the description how equations 1 and 2 are different, both in physical meaning and in mathematical interpretation. Bivariate O-ring statistics should involve two patterns and it's not clear what they are. It's also not explained how ring width and radius interval were determined.

Paragraph starting from line 143: Please define the null model and justify its use. Only afteTar that could one discuss confidence intervals and the expected distribution of point patterns. With Monte Carlo simulation-based confidence intervals, stability should be considered. Also, falling in the envelop does not prove that the distribution is random; it simply means that there's not enough evidence to show deviation from the null model.

Table 2: Please provide the units.

Lines 241 and 242: The term "distribution area ratio" is not defined, and is not immediately obvious in Figure 4. It is unclear how one could reach these conclusions for G40 and G60. Preferential flow paths do not need to be strictly vertical and do not need to be 1-to-1 connected. If there was little vertical flow, how would the dye percolate?

Figures 3-5: individual subplots are too small to be legible and informative to the readers. 

Results and conclusion: This is a major issue. The introduction and the conclusion indicate a rather general and broad research objective, i.e., "the spatial change of preferential flow path under different amounts of precipitation" (line 49-50). However, the entire study was based on a few centimeter-scaled plots, which may not even be representative of the entire Simian Mountain area, not to mention indicative of any general conclusion. The authors should either 1) provide explanation on the limitation of the study and refine the scope of the paper, or 2) justify the representativeness of the experiment to the scope of the study. Without doing so, the conclusion could be fraught with unjustified bold statements that could easily be challenged (for example, line 298: under heavy rain preferential flows are mostly lateral).

Author Response

November 28, 2020

Jin-hua Cheng

School of Soil and Water Conservation,

Beijing Forestry University,

Beijing, China

Dear editor,

Thank you very much for your valuable comments and suggestions with regard to our manuscript entitled “Application of spatial point pattern analysis on preferential flow characteristics under different precipitation amounts” (water-996825), and giving us an opportunity to revise our manuscript.

According to the reviewers’ detailed suggestions, we have revised the manuscript carefully according to all the suggestions and responded to the comments point by point. All revised portions have been marked in red in the revised manuscript, and detailed responses to each comment are presented below. 

Thank you again for your great patience and hard work on our manuscript. If any questions still remain, please notify us. We are looking forward to the positive responses.

Best wishes,

Yours sincerely,

Jin-hua Cheng

Here are our point-by-point responses:

Note: Red text = Editor’s and reviewers’ comments, Blue text = authors’ responses.

Response to Reviewer #1

General comments:

Comments to the Author
This study presents an interesting analysis of the spatial patterns of preferential flow paths with a set of experiment data. The experiment seems overall well-conducted, and the application of the O-ring statistics indeed provided insights in to the experiment data. However, the presentation of the paper needs a lot of improvement to justify the content as a solid scientific journal paper.

Response to general comments:

We greatly appreciate the reviewer’s positive comments. Necessary revisions have been made with respect to the comments of the reviewer, and the responses regarding each of comments list below.

Specific comments:

Comment 1: First of all, I suggest a major revision of the abstract. 1) The audience would not know where Simian mountain or Chongqing province is, so it's better to include a proper reference of a your study area; at lease mention that it's in China (I assumed so). 2) Likewise, the code name (e.g., G20) would be meaningless to the readers of this paper before they actually read the whole paper. Please include proper reference of the experiment sites/setup. 3) The abstract mentions Ripley's K function but the authors actually applied O-ring statistics. This would confuse the audience. 4) In general, the abstract is not well-written in a sense that one could get the overall idea of the paper without reading it. Vague descriptions such as "summation of preferential flow paths" confuses people because the readers do not know how the experiment was conducted, and do not know whether it's summation of cell count, or total size or macropore, etc.

Response: Thanks for the reviewer’s comment. We revised the abstract in light of the above comments. 1) Our study was conducted in Simian mountain grassland of Chongqing province, China. 2) G20 referred to plots which precipitation was 20 mm. Proper reference of the experiment sites has been added in the revised abstract. 3) O-ring statistics was actually applied in our study. 4) “Summation of preferential flow paths” actually referred to the number of preferential flow paths, and has been revised as “The number of preferential flow paths” in the revised manuscript. The abstract has been revised as:

Abstract: Understanding the response of preferential flow paths to water movement is an important topic in soil hydrology. However, quantification of the complicated distribution patterns of preferential flow paths remains poorly understood. Therefore, dye experiments with Brilliant blue FCF were conducted to investigate preferential flow characteristics under three different precipitation amounts (20, 40 and 60 mm, numbered as the G20, G40 and G60, respectively) in Simian Mountain grassland, Chongqing province, China. O-ring statistics was used to analyze spatial distribution characteristics and spatial correlation of preferential flow paths. Results revealed that precipitation could promote dye tracer infiltration into deeper soils, reaching the maximum depth of 55 cm in G60. The number of preferential flow paths in G60 plots was 3.0 and 7.4 times greater than those of G40 and G20, respectively. Structural distribution of the preferential flow paths showed a gradually clumped pattern with the increase of precipitation, which was conducive to enhancing the correlation between preferential flow paths in each pore size range. These results could expand our understanding on the effects of precipitation on the characteristic of preferential flow paths in grassland, which is helpful to evaluate the water movement in the study area.

Comment 2: Line 31: "perfectly predict" is a very strong claim, and I'm not sure any study would claim that there's a way to perfectly predict preferential flow. If the authors believe otherwise, please elaborate.

Response: Thanks for the reviewer’s comment. We are sorry that we did not express it appropriately, and “perfectly” has been removed in the revised manuscript, as shown in line 32.

Comment 3: "Soil Texture": this term is used in line 41 and line 81. I would suggest elaboration of what this term means in this paper's context. soil texture sometimes means different things in the field of geology, hydrogeology, soil mechanics, and foundation engineering. Thus, depending on the background of the audience, one may be very confused by the description of soil texture in line 81.

Response: Thanks for the reviewer’s comment. We are sorry for a vague express. “Soil textural characteristics” in our manuscript actually referred to soil types, and has been revised as “soil types” in the revised manuscript, as shown in line 63, 64 and 95.

Comment 4: Research objective definition: The research objective seems to be in line 49-50. The next paragraph talks about position determination, but the present study does not focus on that part at all. I suggest a reorganization of the last 2 paragraphs in the introduction. It is also not introduced why it's important to study the spatial pattern of preferential flow paths.

Response: Thanks for the reviewer’s comment. We revised the introduction in light of the above comment. We have reorganized all the structure of the introduction. Meanwhile, the importance of studying the spatial pattern about preferential flow paths has been introduced in the first paragraph of the introduction in the revised manuscript, and the sentences about the spatial pattern of preferential flow paths was added in the line 33-41.

Comment 5: Section 2.2: Please specify the precipitation experimental setup. What was the duration, and how was uniform precipitation over the plot achieved, etc. Please specify volumetric or gravimetric moisture content. Also, since double-ring infiltration test is mentioned in the paper, please justify the use of rectangular frames.

Response: Thanks for the reviewer’s comment. Section 2.2 has been revised and Figure 1 has been added in the revised manuscript to introduce the precipitation experimental setup clearly. Meanwhile, the relevant information has been shown in Table 1. Rectangular frames were used in our study instead of double rings according to the study conducted by Sheng Feng et al. [20] to avoid the lateral infiltration of brilliant blue solution, and they have identical effects. Revisions are presented below:

In this study, three rainfall precipitation amounts of 20 mm (moderate rain), 40 mm (heavy rain) and 60 mm (rain storm) were prepared and numbered correspondingly as G20, G40 and G60. Nine plots were established for the three types of rainfall precipitation, with each type having three replicates. In each plot, based primarily on double-ring infiltration, two rectangular steel frames with an inner one of 60 cm (length) × 60 cm (width) × 50 cm (height) and an outer one of 80 cm × 80 cm × 50 cm were concentrically embedded into the soil to a depth of 30 cm. In order to facilitate the subsequent image analysis, we selected a rectangular steel frame instead of a circular one in the experiment, and they have identical effects [20]. After embedding, the soil within 5 cm was compacted to prevent dye tracer infiltration along with the frames. With the use of a sprayer with nozzles (GP1000, Baoding, China) operated above the centre of the frame, brilliant blue FCF (4 g × L-1) was sprayed slowly on the soil surface as a dye tracer to detect preferential flow paths. The dye solution was transported to the sprayer by a constant flow pump (BT100-02, Baoding, China). The usage amount of dye tracer is shown in Table 1. During the spraying experiment, the equal depth of water was sprayed at the soil between two steel frames to avoid the lateral infiltration of brilliant blue solution. After 24 hours, the frames’ soil edges were disturbed when the frames were moved, so only the core part of the soil was analysed. The core region of the dye area (50 × 50 cm) was excavated in 12 horizontal, 5 cm-deep cross-sections. Each horizontal cross-section was vertically photographed with a digital camera (500D, Canon, Japan). The schematic layout of the dyed soil profiles is shown in Figure 1. Next to each plot, we excluded the litter layer and collected soil samples of approximately 500 g in triplicate at 10 cm-deep intervals along the soil depth to measure mass moisture content, bulk density, total porosity, soil mechanical composition, pH and organic matter content (Table 2).

Table 1. . Usage amount of dye tracer

Plot

Spray volume (L)

Spray rate (mmh-1)

Spray duration (h)

G20

7.2

6.35

3.15

G40

14.4

11.98

3.34

G60

21.6

24.19

2.48

Figure 1. Schematic layout of the dyed soil profiles at one sampling position.

Comment 6:Line 118: Please elaborate on the calculation of the path sizes. Please also make it clear how "a path" is determined, and how the corresponding path size is determined. These are all very important aspects of your study and play an important part in the results. Without these details it's difficult for the readers to correctly interpret the results (e.g., how do we interpret "summation of flow path? cell count in the image analysis? total additive path size? path number count?).

Response: Thanks for the reviewer’s comment. As suggested, we have revised the sentence (Line 139- 146) and added a figure of the extraction process of preferential flow paths to help us describe the data treatment process more clearly (Figure 2). The details are as follows:

The Bitmap Analysis function of Image-Pro Plus 6.0 software was used to convert the image into numerical matrices, consisting only two numerical values of 0 and 255. The non-dyed areas composed of many 255 pixel values and the dyed areas composed of many 0 pixel values (Figure 2c). Fourth, the classification and count functions in Image-Pro Plus 6.0 software were used to distinguish the size classes and spatial position messages of the preferential flow in the dyed areas. The watershed function was then set to divide the connected dyed areas into independent individuals. The area of dyed areas was the number of pixels occupied by plaque, which were approximated as rings, and the radius of influence was calculated from the stained area (Figure 2d). According to the back-calculation of the circular area formula, the equivalent radius of the independent preferential paths can be recorded in four classes: 1.0–2.5, 2.5–5.0, 5.0–10.0 and >10.0 mm (Figure 2d). An artificial statistical method was used to get the plant roots’ positions in the horizontal images with the naked eye after the first step of image correction.

Figure 2. Image Analysis.

Comment 7: Line 123: O-ring statistics and K function are actually suited for different research question. The authors should clarify the research question and justify the use of O-ring statistics. It is quite confusing to mention one method in the abstract and introduction, but apply another method in the rest of the study.

Response: Thanks for the reviewer’s comment. We are sorry for a vague express. We have revised the sentence and cited literature to make the description clearer in the revised manuscript. Also, we have revised this vague express in the abstract and introduction. Line 16-17ï¼›154- 161. The details are as follows:

O-ring statistics was used to analyse the spatial distribution characteristics and the spatial correlation of preferential flow paths.

O-ring statistics, based on Ripley’s K-function and the mark correlation function, were used to analyse the spatial patterns. The O-ring statistics replace the circles with rings as a statistical tool to eliminate the bias of accumulative measure using the probability density function.

O-ring statistics include both univariate and bivariate statistics. The statistics were used to analyse the spatial distribution pattern of the preferential flow path with the influence radius (Equation 1). Bivariate statistics were used to analyse the spatial association of the preferential flow paths with two different influence radii (Equation 2). Full details of O-ring statistics processing followed the method described by Thorsten [22].

Comment 8: Equations 1 and 2: Please check for typos as sometimes there's a subscript in the texts but not in the equation. The bigger issue is that these two equations are poorly explained. It's not clear how "n" and "Points[R(r)]" are different based on the description. It is also not clear from the description how equations 1 and 2 are different, both in physical meaning and in mathematical interpretation. Bivariate O-ring statistics should involve two patterns and it's not clear what they are. It's also not explained how ring width and radius interval were determined.

Response: Thanks for the reviewer’s comment. We are very sorry for our carelessness. We have revised equations and cited reference for the full details of O-ring statistics.

,                  (1)

where O11(r) is univariate O-ring statistics, n is the number of preferential flow paths in the observed area, Rw i(r) is the ring with radius r and ring width w centred in the ith point, Points[Rwi(r)] is the number of preferential flow paths within the ring and Area[Rw,i(r)] is the area of the ring.

,               (2)

where Ow12(r) is bivariate O-ring statistics, subscripts 1 and 2 refer to the pattern 1 and pattern 2 respectively, pattern 1 is the pattern with the smaller influence radius, pattern 2 is the pattern with the larger influence radius n1 is the number of preferential flow paths in pattern 1, Rw1,i(r) is the ring with radius r and ring width w centred in the ith point in the pattern 1, Points2[Rw1,i(r)] is the number of preferential flow paths within the ring in the pattern 2 and Area[Rw1,i(r)] is the area of the ring.

Comment 9: Paragraph starting from line 143: Please define the null model and justify its use. Only afteTar that could one discuss confidence intervals and the expected distribution of point patterns. With Monte Carlo simulation-based confidence intervals, stability should be considered. Also, falling in the envelop does not prove that the distribution is random; it simply means that there's not enough evidence to show deviation from the null model.

Response: Thanks for the reviewer’s comment. As suggested, we have defined the null model in line 173-line 180.

Point pattern analysis was performed using Programita 2010 software. It took 1/5 (100 mm) of the sample edge length as the research scale and 1 mm as the step size. To obtain meaningful predictions, we selected different null model separately for the univariate and the bivariate pattern. For univariate point patterns, if the scatterplot was uniformly distributed, then the complete spatial randomness (CSR) null model was used; if the distribution of sample points showed significant spatial heterogeneity, then the heterogeneous poisson process (HPP) null model was used. For bivariate patterns, if the sample points representing the two variables were independent of each other, then the independence null model was used; if influenced by the heterogeneous environment, then the random labeling (RL) null model was used. A total of 19 Monte Carlo simulations were performed to get 95% confidence intervals (i.e. upper and lower package traces).

Comment 10: Table 2: Please provide the units.

Response: Thanks for the reviewer’s comment. As suggested, units had been added in the revised manuscript. Table 3. Root length density (m•m-1)

Comment 11: Lines 241 and 242: The term "distribution area ratio" is not defined, and is not immediately obvious in Figure 4. It is unclear how one could reach these conclusions for G40 and G60. Preferential flow paths do not need to be strictly vertical and do not need to be 1-to-1 connected. If there was little vertical flow, how would the dye percolate?

Response: Thanks for the reviewer’s comment. We are sorry that we did not express it clearly. “Distribution area ratio” refers to the ratio of distribution area of preferential flow paths to total area. We added the values of distribution area ratio in Figure 5. Also, we have revised sentences to help us describe the results more clearly.

Furthermore, Figure 5 demonstrates that a few preferential flow paths existed in the G20, and the distribution of preferential flow paths without apparent various along the depth. Only a few preferential flow paths were generated in the 30 cm layer through the gathering of water and the occurrence of lateral infiltration. In G40, the distribution area ratio of preferential flow paths increased in the 15, 25, 35 and 45 cm layers compared to that in the previous soil layer, showing that the accumulation and lateral infiltration of water flow in these soil layers facilitated the formation and expansion of the preferential flow paths. Additionally, the distribution of the preferential flow paths changed significantly in the 35 to 40 cm soil layers, indicating that the preferential flow paths had a certain degree of lateral offset. In G60, the distribution area ratio of preferential flow paths decreased with increasing soil depth, while the distribution of the preferential flow paths varied insignificantly. This result indicating that the clumped distribution of preferential flow paths increased the vertical transport of water and solutes as well as decreased lateral transport.

 Note: Values in the figure are the distribution area ratio (%). Distribution area ratio is the ratio of distribution area of preferential flow paths to total area.

Figure 5. Distribution map of preferential flow paths in horizon dye profiles

Comment 12: Figures 3-5: individual subplots are too small to be legible and informative to the readers.

Response: Thanks for the reviewer’s comment. As suggested, we have provided the vector images to ensure it can be enlarged and clear.

Comment 13: Results and conclusion: This is a major issue. The introduction and the conclusion indicate a rather general and broad research objective, i.e., "the spatial change of preferential flow path under different amounts of precipitation" (line 49-50). However, the entire study was based on a few centimeter-scaled plots, which may not even be representative of the entire Simian Mountain area, not to mention indicative of any general conclusion. The authors should either 1) provide explanation on the limitation of the study and refine the scope of the paper, or 2) justify the representativeness of the experiment to the scope of the study. Without doing so, the conclusion could be fraught with unjustified bold statements that could easily be challenged (for example, line 298: under heavy rain preferential flows are mostly lateral).

Response: Thanks for the reviewer’s comment. As suggested,We have revised the title as “Study on the preferential flow characteristics under different precipitation amounts in Simian Mountain grassland of China”, and limited the study area throughout the manuscript. The section of “Conclusions” has been revised as:

In summary, this study revealed the responses of spatial distribution and spatial association of preferential flow paths to different water infiltration conditions in the Simian Mountain grasslands of Chongqing, which may be of significance in exploring the characteristics of preferential flow movement. Our results indicated the important contribution of infiltration intensity to the quantities, spatial distribution and spatial association of preferential flow paths. The increase of precipitation amounts was conductive to the preferential infiltration of water into deeper soil. Under moderate rain, preferential flow mainly followed the native preferential flow paths to infiltration and less preferential flow paths were divided. The lateral offset of preferential flow paths were more obvious. Under rain storm, the concentrated distribution pattern resulted in highly continuous and effective preferential flow paths, and the large localised hydraulic gradients could lead to groundwater contamination and subsurface erosion. The findings of this study could provide an reference for future research about the application of preferential flow paths in hydrology processes in the Simian Mountain grasslands. Furthermore, spatial point pattern analysis may be helpful to quantitatively analyse the characteristics of the horizontal spatial distribution of preferential flow paths and thus accurately evaluate the spatial structure characteristics and development degree of preferential flow paths.

Reviewer 2 Report

MINOR REVISION REQUESTED.

In the following, please find enclosed a list of requested minor revisions, mainly referring to the opportunity of reformulating sentences which are not completely understandable to the referee, or to add some comments (and figures) about experimental analysis and set-up.

LINE 84

In chapter 2.2 Experimental design the Authors described the preparation of the sample tests. If it is possible, some pictures (e.g. the iron frames, the sprinklers) may help a lot the reader in understanding experimental set-up and procedures.

LINE 97-100

The A. stated “Soil samples of approximately 500 g were also collected at 10 cm-deep intervals from each plot to measure mass moisture content, bulk density, total porosity, soil mechanical composition, pH and organic matter content” and report the result in Table 1. It is not completely understandable to me, how do they collect the samples (one samples per plot; several samples each plot at a distance of 10 cm each other,…). I suggest reformulating the sentence and to give explicit reference about limiting sediment size (sand, silt, clay). In addition, it would be better to give further details about removing and quantifying organic matter.

LINE 114-115

The sentence is not understandable. Please reformulate

LINE 120-121

It would be helpfull if the A. give few details (or citing reference)  about the artificial statistical method used to get the plant roots’ positions in the horizontal images.

LINE 193-195

The A. refer to investigation into the root systems in the Simian Mountain grassland, about the root’s distribution, reported in Table 2. It is necessary to cite reference or to give further details.

Author Response

November 28, 2020

Jin-hua Cheng

School of Soil and Water Conservation,

Beijing Forestry University,

Beijing, China

Dear editor,

Thank you very much for your valuable comments and suggestions with regard to our manuscript entitled “Application of spatial point pattern analysis on preferential flow characteristics under different precipitation amounts” (water-996825), and giving us an opportunity to revise our manuscript.

According to the reviewers’ detailed suggestions, we have revised the manuscript carefully according to all the suggestions and responded to the comments point by point. All revised portions have been marked in red in the revised manuscript, and detailed responses to each comment are presented below. 

Thank you again for your great patience and hard work on our manuscript. If any questions still remain, please notify us. We are looking forward to the positive responses.

Best wishes,

Yours sincerely,

Jin-hua Cheng

Here are our point-by-point responses:

Note: Red text = Editor’s and reviewers’ comments, Blue text = authors’ responses.

Response to Reviewer #2

General comments:
In the following, please find enclosed a list of requested minor revisions, mainly referring to the opportunity of reformulating sentences which are not completely understandable to the referee, or to add some comments (and figures) about experimental analysis and set-up.
Response to general comments:

Response: Thanks for the reviewer’s comment. We thank the reviewer for the positive comments and greatly appreciate the detailed comments for improving our manuscript. Responses to the reviewer’s concerns in general comments are listed below.

Specific comments:
Comment 1: LINE 84

In chapter 2.2 Experimental design the Authors described the preparation of the sample tests. If it is possible, some pictures (e.g. the iron frames, the sprinklers) may help a lot the reader in understanding experimental set-up and procedures.

Response: Thanks for the reviewer’s comment. As suggested, we made a revision in Figure 1.

Figure 1. Schematic layout of the dyed soil profiles at one sampling position.

Comment 2: LINE 97-100

The A. stated “Soil samples of approximately 500 g were also collected at 10 cm-deep intervals from each plot to measure mass moisture content, bulk density, total porosity, soil mechanical composition, pH and organic matter content” and report the result in Table 1. It is not completely understandable to me, how do they collect the samples (one samples per plot; several samples each plot at a distance of 10 cm each other,…). I suggest reformulating the sentence and to give explicit reference about limiting sediment size (sand, silt, clay). In addition, it would be better to give further details about removing and quantifying organic matter.

Response: Thanks for the reviewer’s comment. Sorry for the confused description. We rephrased that sentence to reflect the information of sample collection. Please see Line 116-119.

Next to each plot, we excluded the litter layer and collected soil samples of approximately 500 g in triplicate at 10 cm-deep intervals along the soil depth to measure mass moisture content, bulk density, total porosity, soil mechanical composition, pH and organic matter content (Table 2).

Comment 4: LINE 114-115

The sentence is not understandable. Please reformulate

Response: Thanks for the reviewer’s comment. Sorry for the confused description. We have revised the sentence (Line 139-146) and added a figure to further explain the Bitmap Analysis function.

The Bitmap Analysis function of Image-Pro Plus 6.0 software was used to convert the image into numerical matrices, consisting only two numerical values of 0 and 255. The non-dyed areas composed of many 255 pixel values and the dyed areas composed of many 0 pixel values (Figure 2c). Fourth, the classification and count functions in Image-Pro Plus 6.0 software were used to distinguish the size classes and spatial position messages of the preferential flow in the dyed areas. The watershed function was then set to divide the connected dyed areas into independent individuals. The area of dyed areas was the number of pixels occupied by plaque, which were approximated as rings, and the radius of influence was calculated from the stained area (Figure 2d).

Figure 2. Image Analysis.

Comment 5: LINE 120-121

It would be helpfull if the A. give few details (or citing reference) about the artificial statistical method used to get the plant roots’ positions in the horizontal images.

Response: Thanks for the reviewer’s comment. We revised the sentence “An artificial statistical method was used to get the plant roots’ positions in the horizontal images with the naked eye after the first step of image correction.” Line148-150.

Comment 6: LINE 193-195

The A. refer to investigation into the root systems in the Simian Mountain grassland, about the root’s distribution, reported in Table 2. It is necessary to cite reference or to give further details.

Response: Thanks for the reviewer’s comment. As suggested, table 3 has been revised.

Table 3. Root length density (mm-1)

Soil depth (cm)

Root diameter

< 1 mm

1–3 mm

3–5 mm

5–10 mm

>10 mm

0–5

907.33±54.02

166.93±49.83

186.93±121.08

38±27.94

4±3.27

5–10

179.87±50.43

25.33±22.31

3.87±5.47

12±8.64

2.83±4

10–15

94.56±59.12

35.12±21.61

1.39±1.96

3.97±5.62

0

15–20

100.88±26.81

0

6.24±6.54

0±0

0

20–25

48.19±13.67

6.21±7.79

0

1.04±1.47

0

25–30

41.2±9.41

3.73±5.28

0

1.33±1.89

0

30–35

57.65±60.58

32.8±41.4

2.72±2.01

0.69±0.98

0

35–40

124.45±97.23

41.87±32.77

23.81±26.16

13.6±10.79

0

40–45

40.29±20.69

22.32±20.97

10.37±11

3.89±2.76

0.72±1.02

45–50

14.4±10.37

1.07±1.51

0

5.6±2.26

0.69±0.49

50–55

8.77±9.22

1.63±1.34

0

3.41±1.28

0

Total root length density

1617.59

337.01

235.33

83.53

8.24

Round 2

Reviewer 1 Report

The authors did a nice job revising the manuscript and made significant improvements.

A few comments were not fully addressed and thus it'd be great if the authors could further refine the manuscript accordingly.

Equations 1 and 2: The explanation of the equation has been improved, but the influence radius of the preferential flow path was not clearly explained (e.g., whether that's determined via the image analysis, and how, etc.). Is that the same as "pore size range" shown in Figure6? It in unclear also because in Figure 2 it appears that there are only two cell values, 0 and 255. The first paragraph in section 3.3 mentions "spatial location relationships between different soil preferential flow paths", but it's unclear what the difference is.

Figures 4-6: It's great that the authors provided vector images, but I think those were rendered as raster images in the pdf file. As a result, individual subplots are still quite illegible.

Finally, a few typos are in the added texts. Please proofread and correct them.

Author Response

December 06, 2020

Jin-hua Cheng

School of Soil and Water Conservation,

Beijing Forestry University,

Beijing, China

Dear editor,

Thank you very much for your valuable comments and suggestions about our manuscript entitled “Application of spatial point pattern analysis on preferential flow characteristics under different precipitation amounts” (water-996825), and giving us an opportunity to revise our manuscript.

According to the reviewers’ detailed suggestions, we have revised the manuscript carefully according to all the suggestions and responded to the comments point by point. All revised portions have been marked in blue in the revised manuscript, and detailed responses to each comment are presented below.

Thank you again for your great patience and hard work on our manuscript. If any questions remain, please notify us. We are looking forward to positive responses.

Best wishes,

Yours sincerely,

Jin-hua Cheng

Here are our point-by-point responses:

Note: Red text = Editor’s and reviewers’ comments, Blue text = authors’ responses.

Response to Reviewer

General comments:

Comments to the Author
The authors did a nice job revising the manuscript and made significant improvements.

A few comments were not fully addressed and thus it'd be great if the authors could further refine the manuscript accordingly

Response to general comments:

We greatly appreciate the reviewer’s positive comments. Necessary revisions have been made concerning the comments of the reviewer, and the responses regarding each of the comments list below.

Specific comments:

Comment 1: Equations 1 and 2: The explanation of the equation has been improved, but the influence radius of the preferential flow path was not clearly explained (e.g., whether that's determined via the image analysis, and how, etc.). Is that the same as "pore size range" shown in Figure6? It in unclear also because in Figure 2 it appears that there are only two cell values, 0 and 255. The first paragraph in section 3.3 mentions "spatial location relationships between different soil preferential flow paths", but it's unclear what the difference is.

Response: Thanks for the reviewer’s comment. We are sorry that we did not express it appropriately.

(1) We have revised the sentence (Line 144-148; 172-173) and fixed Figure 2.

(2) The pore sizes (the influence radius) covered in this article were obtained from image analysis.

(3) The values 0 and 255 represented black and white, i.e., stained and unstained areas.

(4) “spatial location relationships between different soil preferential flow paths” actually referred to preferential flow paths for different pore sizes ranges, and has been revised as “To further reveal the spatial structure characteristics of soil preferential flow paths, spatial correlation analyses of the spatial location relationships between preferential flow paths for different pore sizes ranges were conducted”(Line 294-295).

The details are as follows:

Fourth, the classification and count functions in Image-Pro Plus 6.0 software were used to distinguish the size classes and spatial position messages of the preferential flow in the dyed areas. The watershed function was set to divide the connected dyed areas into independent individuals (Figure 2d). The area of each dyed area was the number of pixels occupied by plaque, which were approximated as rings, and the radius of influence was calculated based on the stained area (Figure 2e). Then, a data file was generated containing the radius of dyed areas and the coordinates at the center point of the dyed areas (Figure 2d). This study classified radii of preferential paths into four classes: 1.0–2.5, 2.5–5.0, 5.0–10.0 and >10.0 mm.

Point pattern analysis was performed using Programita 2010 software. The initial input data was a DAT file exported by image analysis, containing the coordinates and radii of the preferential flow paths. It took 1/5 (100 mm) of the sample edge length as the research scale and 1 mm as the step size.

Figure 2. Image Analysis.

Comment 2: Figures 4-6: It's great that the authors provided vector images, but I think those were rendered as raster images in the pdf file. As a result, individual subplots are still quite illegible.

Response: Thanks for the reviewer’s comment. As suggested, we have split Figure 4 (Figure 6)into three figures showing the O11(r) (O12(r)) values for G20, G40 and G60, respectively. The split diagram has provided in the supplementary material (Figure S1; Figure S2). For Figure 5, since there is no overlap, we believe that the clarity of the vector graphics will not bother the reader.

Figure S1. Change of horizontal distribution pattern function O11(r) with scale at different soil depths

Figure S2. Change of spatial association function O12(r) with scale at different soil depths

Comment 3: Finally, a few typos are in the added texts. Please proofread and correct them

Response: Thanks for the reviewer’s comment. We are sorry for the typos. We have revised it throughout the manuscript. All revised portions have been marked in blue in the revised manuscript.
